# Main Characteristics of Processed Grain Starch Products and Physicochemical Features of the Starches from Maize (*Zea mays* L.) with Different Genotypes

**DOI:** 10.3390/polym15081976

**Published:** 2023-04-21

**Authors:** Eduard B. Khatefov, Vladimir G. Goldstein, Alexey V. Krivandin, Lyubov A. Wasserman

**Affiliations:** 1N.I. Vavilov All-Russian Institute of Plant Genetic Resources (VIR), 42-44, Bolshaya Morskaya Street, 190000 St Petersburg, Russia; 2Branch of Russian Potato Research Centre (ARRISP-RPRC), All-Russian Research Institute of Starch and Starch-Containing Raw Materials Processing, 11, Nekrasova Street, 140051 Kraskovo, Russia; 6919486@mail.ru; 3Emanuel Institute of Biochemical Physics RAS (IBCP RAS), 4, Kosygina Street, 119334 Moscow, Russia; a.krivandin@sky.chph.ras.ru (A.V.K.); lwasserma@mail.ru (L.A.W.)

**Keywords:** *Zea mays* L., diploid and tetraploid genotypes, grain characteristics, starch, amylose, morphological, structural and thermodynamic properties

## Abstract

To understand the relationship between the genotype of maize plants and differences in their origin and the ploidy of the genome, which carry gene alleles programming the biosynthesis of various starch modifications, the thermodynamic and morphological features of starches from the grains of these plants have been studied. This study investigated the peculiarities of starch extracted from subspecies of maize (the dry matter mass (DM) fraction, starch content in grain DM, ash content in grain DM, and amylose content in starch) belonging to different genotypes within the framework of the program for the investigation of polymorphism of the world collection of plant genetic resources VIR. Among the starch genotypes of maize studied, four groups comprised the waxy (*wx*), conditionally high amylose (“*ae*”), sugar (*su*), and wild (WT) genotypes. Starches with an amylose content of over 30% conditionally belonged to the “*ae*” genotype. The starches of the *su* genotype had fewer starch granules than other investigated genotypes. An increase in amylose content in the investigated starches, accompanied by a decrease in their thermodynamic melting parameters, induced the accumulation of defective structures in the starches under study. The thermodynamic parameters evaluated for dissociation of the amylose–lipid complex were temperature (*T_aml_*) and enthalpy (*H_aml_*); for the *su* genotype, temperature and enthalpy values of dissociation of the amylose–lipid complex were higher than in the starches from the “*ae*” and WT genotypes. This study has shown that the amylose content in starch and the individual features of the maize genotype determine the thermodynamic melting parameters of the starches under study.

## 1. Introduction

Cereal crops are the primary sources of starch. In addition, tuber crops (e.g., potato, manioc, and others), legumes, and some unripe fruits, such as bananas and mangoes, contain starches [1]. As known, starches consist of two polysaccharides—linear amylose and branched amylopectin; some starches also consist only of amylopectin molecules. Both polysaccharides are homopolymers of α-D-glucopyranose (glucose), differing in molecular mass and physicochemical properties. A linear amylose molecule consists of α-glucose residues linked with α-(1→4)-glycoside bonds. Branched chains of α-glucose residues connected with α-(1→4) and, in the chain branching points, α-(1→6) glycosidic linkages form amylopectin macromolecule. The structure of amylopectin is three-dimensional: its branches point in all directions, thus contributing to the spherical shape of its molecule [2,3]. The ratio of the two main polysaccharides in starches determines their functional and physicochemical properties. The ratio of amylose to amylopectin in starch is responsible for its rheological properties, whereas starch swelling power and solubility depend on the interaction among polymer chains, incorporating amorphous and crystalline fractions of starch granules [4]. The ratio of amylose to amylopectin determines the degree of this interaction and depends on the degree of polymerization, chain length, branching degree, molecular mass, and molecular configuration [5]. Starch swelling power is directly linked to amylopectin content because amylose acts as a diluent and swelling inhibitor [6,7]. In the large-scale production of native starches, the ratio of amylose to amylopectin averages 1:3. The ratio of amylose to amylopectin in starches varies by maize subspecies but is typically in the range of 20–30% in normal cereal starches. Various plant genetic mutations form three main classifications of starches (waxy, normal, and high amylose) [8]. Normal maize starches contain 24–28% amylose; waxy maize starches in different variants contain amylose of less than 10% [9]. The amylose content in high-amylose maize starches was more than 50% [10,11]. Mutant lines of maize are currently known, where starch contains about 85% amylose [12]. Applying waxy, normal, and high amylose maize starches is different. Waxy and normal maize starches are usually used as fillers, thickeners, or stabilizers in the food industry [13]. High-amylose starches are used in the production of preventive foods as dietary fibers [10], emulsion stabilizers [8], biodegradable plastics [14], and pharmaceutical materials [15]. The rheological properties of maize starches with increased amylose content differ significantly from the conventional maize starches to those produced from waxy maize [2]. High melting temperatures of crystalline lamellae [16], high swelling power, viscosity, the durability of their colloids during the pasting process, and the ability to impede amylose retrogradation, compared to conventional and high-amylose starches, characterize amylopectin maize starches.

Structural and thermodynamic properties of starch were shown to be affected by the conditions of starch-yielding plant cultivation and by the presence of specific genes involved in the biosynthesis of starch polysaccharides [17]. The activity of various enzymes involved in the biosynthesis of starch polysaccharides determines the ratio of the main polysaccharides in starch and their physicochemical properties. The synthases responsible for the synthesis of starch and glycogen polysaccharides, represented by starch synthase (SS), glycogen synthase (GS), and starch phosphorylase (SP), belong to the glycosyltransferase-B (GT-B) superfamily (CL0113) [18]. Five independent conservative SS (GBSSI, SSI, SSII, SSIII и SSIV) have been found in plants [19]. The granule-bound starch synthase (*GBSS*) is crucial for amylose synthesis [20], while mutations in SSIIb and SSIII alter the levels of amylopectin content [21]. Three known forms of expressed enzymes promote the branching of starch molecules in the endosperm of cereals with varying degrees: enzyme I (SBEI), enzyme IIa (SBEIIa), and enzyme IIb (SBEIIb) [22]. The genotypes with the waxy type of grain endosperm (*wx*) are characterized by the absence of GBSS accompanied by the accumulation of only amylopectin in the complete absence of amylose. The amylose type of grain endosperm (*ae*) is characterized by a violation of SBEIIb accompanied by the effect of the accumulation of weakly branched polysaccharides and the accumulation of amylose in starch [23,24]. The maize with a very high level of amylose (>90%) was obtained in considerably reduced enzymatic activity of SBEI and almost suppressed activity of SBEII [25,26]. Mutant SBEIIb genotypes form starch granules in the endosperm with abnormal structure morphology. An increase in amylose content increases defects in partially crystalline starch granules that decrease starch melting temperature. In addition, this genotype contains a considerable amount of a substance of the intermediate type between amylose and amylopectin [8,27]. The sugary (*su*) genotype with sucrose synthases (SS) generating phytoglycogen as an amylopectin precursor has a deficit of such enzymes as isoamylases and pullulanases disturbing branching in starches. The high sugar and low starch contents in sweet sorghum and maize indicate the high activity of this enzyme. Grains of these plants, as a rule, are wrinkled with a low DM concentration [28,29,30]. In sweet maize, with an increase in the number of chromosomes from diploid (2*n*) to tetraploid (4*n*), the number of alleles of the *su*2 genes that control the synthesis and accumulation of sugars in the endosperm of grain with the genotype *su2su2su2* (2*n*) *to su2su2su2su2su2su2* (4*n*) is increasing. Accordingly, the number of dominant alleles of the *Su2* gene (wild type) in ordinary corn will affect the synthesis of normal starch in the same proportions as *Su2Su2Su2* (2*n*) *Su2Su2Su2Su2Su2Su2* (4*n*) without the accumulation of sugar in the maize grain. The findings of investigations into the grain chemical components affecting the taste qualities of tetraploid and diploid sweet maize have shown that an increase in the number of genes is accompanied by a change in the quantitative, rather than qualitative, biochemical profile characteristics [31]. Studies of the effect of different genome ploidy of waxy or amylose maize on starch properties have not been found in the literature.

Our study of polymorphism of plant genetic resources of the world VIR maize collection determined standard values of thermodynamic and structural features of starches isolated from maize genotypes of different origins and genomic ploidy, carrying alleles of genes programming biosynthesis of various starch modifications. Tetraploid genotypes found in the standard study of the diversity of the VIR world maize collection resources allowed a comparative analysis of some starch properties with diploid ones. The prospect of possible regulation of starch quality by changing the number and ratio of different genes responsible for starch synthesis is a promising direction in genetics and breeding. It is also possible to adjust the granule morphology and crystallinity of extracted starches at the genetic level. As a rule, not only one or two genes but also a gene network regulates starch morphological features, including its physicochemical properties. Thus, the genotypes containing genes determining the starch structure and physicochemical properties will differ in all or some parameters, representing optimal raw materials for different industries. In this regard, the purpose of this study was to evaluate the characteristics of the products of processing grain for starch, as well as to understand the relationship between the thermodynamic and structural features of maize starches from samples of the VIR world collection with different ploidy and genotype of maize plants from which starches are extracted.

## 2. Experimental Section

### 2.1. Materials

The maize material was reproduced for the research in 2017 in a foothill area at the Kabardino-Balkarian Research Institute of Agriculture. The breeding site was within the piedmont zone of the North Caucasus, at the watershed dividing the rivers Urvan and Nalchik. Meadow chernozems mainly represented soils. Cultivars and hybrids were reproduced under paper cages using the technique of cross-pollination within a population on a plot. Harvesting was manual, scheduled as soon as the ears reached full ripeness. The maize seed material was represented by 11 accessions of non-transgenic flint, dent, and sweet maize subspecies from the global collection held by the N.I. Vavilov All-Russian Institute of Plant Genetic Resources (VIR, St. Petersburg, Russia), selected for their high starch content in grain as part of a program for the study of polymorphism of plant genetic resources of the VIR world collection. Among the selected accessions were cultivars (Kabardinskaya belaya zubovidnaya, White Flint, Gornaya chechenskaya, Mais agestano, Populyatsiya MRPP22, Baksanskaya sakharnaya and Luch, Alina, Ranyaya Lacomka, Mestnaya) of Russian and foreign origin and a hybrid (Otbornyy 150SV) developed domestically. The accession Baksanskaya sakharnaya was isolated from Populyatsiya MRPP22 as a mutant for the *su*2 gene. Tetraploid maize (2*n* = 40) (Baksanskaya sakharnaya, Populyatsiya MRPP22) are closely related genotypes that differ from each other only by a mutation in the *su*2 gene, from diploid maize (2*n* = 20) by a double set of chromosomes, and a great size grains and germ (Figure 1 and Figure 2). In our study, all samples were divided into amylopectin (*wx*), sugar (*su*), WT (wild), and conditionally high amylose (“*ae*”). Maize samples with an increased amylose content of more than 30% (varieties White Flint, Gornaya chechenskaya) in our study are conditionally assigned to subspecies of amylose type “*ae*” since the increased content of amylose starch does not allow us to classify these samples as normal, but they are not yet typically high amylose.

### 2.2. Methods

#### 2.2.1. Cytological Analysis

In the morning hours, specimens of maize seedling rootlets prepared for chromosome analysis were fixed in acetalcohol (Carney’s fixative solution) in the phase of three-day-old plantlets. Chromosome staining was performed according to the Feulgen protocol, using the Schiff reagent after hot hydrolysis in 1N HCL at 60 °C for 10 min; beyond that point, chromosomes were stained at room temperature for 1 h with the subsequent triple washing of specimens in sulfur water [32]. Maceration with cellulase from *Aspergillus niger* (Sigma-Aldrich, Burlington, MA, USA) was employed to facilitate the crashing of the specimens.

#### 2.2.2. Extraction of Starch and By-Products

Starch was isolated from maize grain using the modified technique offered by Adkins and Greenwood [33]. Maize grain (100 g) was immersed in a 0.4% solution of sodium metabisulphite (initial bulk concentration of SO_2_ was ~0.20%) and infused for 48 h at 48–50 °C. The maize extract was separated from the grain (coarsely crushed in a PLU-1 laboratory crusher (All-Russian Research Institute of Starch and Starch-Containing Raw Materials Processing, Russia)). The ground grain was thoroughly washed from the crusher with 100 mL of distilled water. The maize germ was isolated from the grain matter, rinsed to remove free starch through a 0.5 mm metal sieve mesh with 100 mL of water, and dried in a drying cabinet at 50–52 °C. After that, the grain matter separated from the germ underwent fine grinding in a Waring LB20ES laboratory blender (Conair Group, Cranberry Township, PA, USA) at 40 °C for 5 min with a rotational speed of 3000 rpm. The resulting fine substance underwent sifting through six 70 μm nylon sieve meshes. The grain matter left on the sieve surface underwent multiple washing with distilled water (9 times, using 100 mL of water for each washing) until the rinsing water had become starch-free, verified by the iodine test. The resulting cellulose contained different mass fractions of the ‘bound’ starch and then dried at 55–58 °C until the constant weight. The starch/protein suspension underwent centrifugation (Hermle Labortechnik, Z300K, Wehingen, Germany) for 30 min at 6000 rpm, and the starch and protein (corn gluten) sediment underwent further separating by washing; repeated centrifugation separated the gluten layer from the starch. The gluten suspension separated from starch was centrifuged (Hermle Labortechnik, Z300K, Wehingen, Germany) for 20 min at 6000 rpm. The obtained starch and gluten were dried separately at 50–55 °C for 24 h with subsequent measuring of the product yield and the mass fraction of starch in by-products, including the same in the rinsing water. The resulting starch was studied using the described physicochemical methods.

#### 2.2.3. Assessment of Grain Characteristics

Grain characteristics (moisture, ash, and starch content) were assessed in compliance with the standard European techniques (ISO 6540:1980: Maize—Determination of moisture content (on milled grains and whole grains]; ISO 2171:2007: Cereals, pulses, and by-products—Determination of ash yield by incineration; ISO 10520:1997: Native starch—Determination of starch content—Ewers polarimetric method).

#### 2.2.4. Measuring the Amylose Content

Amylose content was measured colorimetrically according to the described technique [34]. The result was compared with the Megazyme amylose/amylopectin assay procedure, utilizing the commercial kit (Megazyme Ireland International, Ltd., Bray, Ireland), followed in compliance with the manufacturer’s recommendations.

#### 2.2.5. Light Microscopy

Chromosome numbers of the specimens were calculated using transmitted light microscopy. For each maize accession, the number of chromosomes in a somatic cell from the rootlet was counted in 15 metaphase plates of the crushed specimen under an Olympus CX43 microscope (Olympus Optical Co., Ltd., Osaka, Japan) using 1600× zoom with immersion.

Light microscopy was used to analyze the morphology of the starch granules under study. Starch was placed on a microscope slide and stained with one drop of Lugol’s iodine (a solution of potassium iodide with iodine in water) [35]. The preparation was then covered with glass to ensure an even distribution of starch granules under the glass. The filter paper removed the excess dye. Starch granules were analyzed using 400× zoom and a blue filter under a Micromed 3 lm optical microscope in transmitted light with an attached Oplenic PSC-600-15C (B51) imaging camera (Oplenic Corp., HangZhou, China).

#### 2.2.6. Scanning Electron Microscopy (SEM)

Morphological changes of the starch granule were investigated with scanning electron microscopy (SEM), obtaining photos of starch granules using a scanning electron microscope Mira3 LMU (Tescan, Brno, Czech Republic) at room temperature in the conditions of a high vacuum with an accelerating voltage of 500 V.

The size and morphology of the maize starch granules were measured or estimated using an extended suite of morphometric algorithms under the “Altami Studio” licensed software GUI. Besides the standard granulometric characteristics, the ellipticity factor was analyzed and compared. The ellipticity factor was calculated using the abovementioned software as the ratio between the starch granule squares and squares of the ellipses with equivalent moments of inertia, taking the ellipticity coefficient to be unity in the case of elliptic approximation.

#### 2.2.7. X-ray Diffraction

X-ray diffraction (XRD) study of maize starches was carried out in an air-dry state using the modernized X-ray diffractometer HZG4 (Freiberger Präzisionsmechanik, Freiberg, Germany) operated in Bregg-Brentano geometry with diffracted beam graphite monochromator and scintillation counter (CuKα radiation). X-ray diffraction patterns for each starch sample were recorded three times and then averaged. The total starch crystallinity *C* was calculated as in [36] with the equation
(1)$$C=\frac{\int_{2\theta_{1}}^{2\theta_{2}}\left[I\left(2\theta \right)-I_{b}\left(2\theta \right)-I_{a}\left(2\theta \right)\right]d\left(2\theta \right)}{\int_{2\theta_{1}}^{2\theta_{2}}\left[I\left(2\theta \right)-I_{b}\left(2\theta \right)\right]d\left(2\theta \right)}\cdot 100\%,$$
carrying out intensity integration in 2*θ* range from 6° to 36°. In this equation, *I*(2*θ*) is experimental XRD intensity of a starch sample, *I_b_*(2*θ*) is a baseline intensity which is assumed to be represented by a straight line passing through experimental intensity points at 2*θ* ≈ 6°, and *I_a_*(2*θ*) is the hypothetical intensity from the starch amorphous part. *I_a_*(2*θ*) was determined on the basis of experimental XRD pattern of amorphous starch obtained by milling of a normal maize starch (amylose content 20–22%) in a ball mill for 8 h [37]. For calculation of a starch sample crystallinity with Equation (1), experimental XRD intensity of amorphous starch was diminished for each starch sample in such a way that this intensity was approximately equal to the intensity of this sample at 2*θ* ≈ 21°.

#### 2.2.8. Differential Scanning Calorimetry

Thermodynamic melting parameters for 0.3% (*w*/*w*) water dispersions of the investigated maize starches were measured using the method of highly sensitive differential scanning calorimetry (DSC) on a DASM-4 microcalorimeter (Pushchino, Russia). The volume of the sample under study was 0.5 cm^3^ in a closed capsule. Measurement conditions were in the temperature range of 20–120 °C, at the heating rate of 2 °C/min, and under the constant pressure of 2.5 atm, repeating DSC measurements for each sample three times. The excess heat capacity scale was calibrated using the Joule–Lenz effect for each experiment. As shown earlier, under the given conditions, there was no need to consider thermal lags or the sample treatment duration in the calorimeter capsule [38]. Deionized water (Millipore Direct-Q3) with a special resistance of 18.2 MΩ at 25 °C was used for comparison as the standard solution. 

Mean values of thermodynamic melting parameters for the crystalline lamellae were identified in starch out of no less than three concurrent measurements. The melting temperature value matched the maximum heat capacity peak on the thermogram. The value of experimental enthalpy corresponded to the area under the peak of the excess heat capacity curve as a function of temperature. Molar enthalpy of melting (ΔH_m_) was calculated for the anhydroglucose residue (162 g/mol). As a rough approximation, the starch melting process can be considered a quasi-equilibrium one [39,40], which opens an opportunity to apply a one-stage melting model where the starch melting process is an equilibrium reaction between the native and melted states. 

The Van’t Hoff enthalpy values (ΔH^νH^) were calculated as described earlier [40] using the following equation:ΔH^νH^ = 2 R^1/2^ T_m_ (C_p_ − 0.5ΔC_p exp_) ^½^(2)
where R is the universal gas constant; T_m_ is the melting temperature for a crystalline lamella in starch; C_p_ is the maximum on the ordinate of the heat capacity peak on the thermogram; ΔC_p exp_ is the difference in heat capacity values between native and molten state. The cooperative melting unit (ν) and thickness of the crystalline lamella (L_crl_) were calculated as described in the published source [41].
*ν* = (ΔH^vH^)/(Δ*H*_m_)(3)
where ΔH_m_ is the experimental enthalpy of melting for the crystalline lamella and ΔH^vH^ is the Van’t Hoff enthalpy of melting.
L_crl_ = 0.35ν(4)
considering that 0.35 nm is the projection of the anhydroglucose residue on the axis of the amylopectin double helix [42,43].

The thermodynamic melting parameters of crystalline lamellae, namely the surface entropy (q_i_), the free surface energy (γ_i_), and enthalpy (s_i_) for the faces of crystalline lamellae, are possible to obtain from the Gibbs–Thomson equation for semicrystalline polymers [44].
T_m_ = T^0^_m_ (1 − 2γ_i_/(ΔH^0^_m_ ρ_crl_ L_crl_))(5)
where T^0^_m_ and ΔH^0^_m_ are the melting temperature and enthalpy, respectively, of a hypothetic crystal of unlimited size (ideal crystallite) or such crystal whose role of free surface energy may be ignored in favor of bulk energy; γ_i_ is the free surface energy for the faces of crystalline lamellae; p_crl_ and L_crl_ are the density and thickness of a crystalline lamella, respectively.
q_i_ = (ΔH^0^_m_ − ΔH_m_) ρ_crl_ L_crl_/2.5(6)
*s*_i_ = (q_i_ − γ_i_)/T_m_(7)

#### 2.2.9. Statistical Analysis

The data were reported as an average of triplicate observations and subjected to statistical analysis by Microcal Origin.

## 3. Results and Discussion

### 3.1. Ranking Maize Accessions According to the Type of Starch and Maize Grain Characteristics

Studying the ranking of the accessions according to their grain starch genotypes and analyzing the results helped to identify the accessions belonging to the *wx* (Mais agestano, Mestnaya), conditionally to “*ae*” (White Flint and Gornaya chechenskaya) and *su* types (Baksanskaya sakharnaya, Alina, Ranyaya Lacomka) (Table 1). The conditionality of attributing some (White Flint and Gornaya chechenskaya) samples to type “*ae*” is based on the fact that the content of amylose starch was higher (above 30%) than that of ordinary corn (up to 25%), but did not reach the value of high-amylose maize (above 50%). The remaining maize samples (Luch, Population MRPP22, Kabardinskaya belaya zubovidnaya, Otborny 150CB) are classified as wild *(*WT*)* types based on the starch content in the grain corresponding to the content in ordinary maize. The cytological analysis resulted in the finding that accessions Baksanskaya sakharnaya and Populyatsiya MRPP22 have the tetraploid chromosome number (4*n*), whereas all other accessions have the diploid (2*n*) number of chromosomes in their genome (Table 2). Tetraploid accessions differ from diploid ones in that the latter have a twofold set of all genes in their genome, and tetraploid ones have a fourfold *sugary2* gene in the homozygous recessive *su2su2su2su2* (Baksanskaya sakharnaya) and dominant *Su2Su2Su2Su2* (Populyatsiya MRPP22) states. Grains carrying alleles of the *su2*, *wx*, *ae* gene in the genome showed slight phenotypic differences mainly pronounced in sweet maize (Figure 3I).

Table 1 presents the genetic characteristics of the studied maize grains, their starch content, and amylose content in starch. The data made it clear that the maize accessions with the amylopectin genotype (*wx*) and those with the amylose one (“*ae*”) demonstrated similar values of the DM mass fraction; with this, the amylopectin genotype possessed higher amounts of minerals compared with the amylose genotype reflected in the mass fraction level of ash content in grain. Both genotypes had close to similar levels of starch content in grain: from 69.3 to 73.7%. Here, the starches from the amylose genotype contained 32.0–38.8% of amylose, and those from the amylopectin genotype 0–15% of amylose. The sugary genotype (*su*), represented by tetraploid accession, showed the highest value of the DM mass fraction (from 92.8% up to 93.3%), the lowest starch mass fraction (58.8–65.9%) in grain DM, and the ash mass fraction of 1.5–1.7% in grain DM. Concurrently, the minimum mass fraction of starch from grains of *su* genotypes was accompanied by the amylose fraction in starch equal to 25.5–27.7% of grain DM. These data confirmed the assumption that the number of the main polysaccharides in starch (amylose and amylopectin) and their ratios are determined by the plant genotype. Nevertheless, there were some differences in the studied parameters within each of the four groups with the sugary, amylose, amylopectin, and wild endosperm types.

Maize accessions with the *wx*, “*ae*,” and WT genotypes showed high starch content after profound grain processing (from 69.3 to 75.3% of dry matter). The content of the *su* genotype starches in the grains with corresponding genotypes was high enough for sweet maize, reaching 59.3–65.9% of the grain dry matter, depending on the variety.

The results of starch extraction from the grain with different maize genotypes during laboratory-based processing (Table 1) demonstrated the effectiveness of starch extraction from the grain of the “*ae*” maize genotype. Starch extraction from the tested maize cultivars White Flint and Gornaya chechenskaya showed an 87–90% ratio, i.e., more than other studied accessions with different genotypes. However, grain samples of the *wx* type had a starch extraction ratio of 83.1–84.6%, which is high enough for the processed grain of waxy maize. Note that the starch yield from grains of the *su* genotype was the smallest among the investigated maize cultivars. Little starch granules, non-typical for maize grains, explained a low ratio of starch extraction from the *su*-type grains (Baksanskaya sakharnaya, Alina, Ranyaya Lacomka), so in this case, the losses in by-products tended to increase significantly.

The mass fraction of ash in the grain samples under study with the *wx* genotype did not change significantly and amounted to 1.4–1.7% of dry matter. The mass fraction of amylopectin in starch from maize with the *wx* genotype was close to 100%.

The highest mass fraction of amylose in starch among the investigated maize starches was for starch of the “*ae*” genotype (varieties Gornaya Chechenskaya and White Flint, 32% and 38%, respectively). Among the cultivars with the *su* genotype, the mass fraction of amylose in starch in variety Ranyaya Lacomka was 27.7%, the highest in comparison with other investigated varieties with the *su* genotype.

Thus, the lowest starch content in the grains of the *su* genotype was observed, and the highest starch content was in the grains of the wild genotype with the following set, depending on the plant genotype for the increase of amylose content in starch: *wx* ≤ WT ≤ *su* ≤ “*ae*”.

### 3.2. Analysis and Characteristics of Starch Granule Size

Figure 3II presents the results of a microscopic examination performed on the starch granules from different genotypes in the presence of Lugol’s solution. Amylopectin and amylose molecules were stained with Lugol’s solution differently. Amylopectin macromolecules turned brown, and amylose macromolecules turned blue [35]. The microscopic images showed that brown amylopectin molecules were mainly present in the *wx*-type starch, but the “*ae*” images and *su* genotypes contained both blue macromolecules of amylose and brown ones of amylopectin. These micrograph images also demonstrated that the starches isolated from different maize genotypes had different granule sizes. It is shown that the granules of the large and small starch fractions contained different quantities of amylose and amylopectin molecules. According to the published data, the intensity of crystalline reflexes is higher in starch samples with larger granules than in those with smaller granules. Therefore, these factors may affect the structural arrangement of starch granules and the thermodynamic parameters of starch melting [45].

The starch morphology in the granules was analyzed by scanning electron microscopy (SEM). The size and shape of maize starch granules were evaluated both using direct morphometric parameters and relative values (the ellipticity coefficient (calculated as the ratio of the area of the object to the area of the ellipse with the same moments of inertia)). Figure 3III presents examples of micrographs of maize starch granules of different genotypes. The microphotographs showed that all studied starches contained both irregular or cubic granules and elliptical granules with clear edges, the ratio of which differed in different genotypes.

Note the different sizes of maize starch granules from plants with different genotypes (Figure 4). Thus, from the above morphological data, large geometric dimensions compared to the starch granules of the *wx* genotype characterized the maize starch granules of the “*ae*” genotype, while the proportion of irregular or cubic granules in the starch of the “*ae*” genotype was higher than in the starch of the *wx* genotype. The size of the starch granules WT genotype had higher values of geometric parameters compared to the *wx* genotype of starch granules and lower values of the same parameters than the “*ae*” genotype. Simultaneously, the proportion of irregular granules in the starches WT genotype was lower than in the “*ae*” genotype of starch but slightly more than in starches of the *wx* genotype. It should be emphasized that starch granules of the *su* genotype had significantly smaller sizes compared to granules of the “*ae*” and *wx* genotypes, which may be due to the individual genotypic features of this genotype. Hence, the following settings could be drawn up according to the increase of geometric parameters for investigated starches depending on the plant genotype: *su* ≤ *wx* ≤ WT ≤ “*ae*”. Note that the starch of *wx* type had a lower proportion of irregular granules, and it is possible to imagine the increase in the number of irregular starch granules in the investigated starches dependently on the plant genotype as follows: “*ae*” ≥ *su* ≥ WT ≥ *wx*. Note that our results on the granule size of the *su* maize genotype correlated with the literature data [46]. It is possible that the size of maize starch granules can serve as a phenotypic marker for distinguishing starches of the *su* type from those isolated from the *wx* and “*ae*” genotypes.

### 3.3. X-ray Diffraction

Figure 5 shows the XRD patterns of maize starches of various genotypes. As known, A-polymorphous structure is characteristic of waxy (*wx*-genotype) and normal maize (transitional from *wx* to *ae* genotype) starches [11] and B-polymorphous structure—for high-amylose maize starches (*ae*-genotype) [45,46]. It can be seen that all investigated starches showed peaks at 2θ ≈ 15, 17, 18, and 23°, typical for A-type polymorphous structures. Hence, our investigated starches have an A-type polymorphous structure correlated with literature data for waxy, normal, and sugar maize starches [11,46,47]. The intensity of the peak at 2θ ≈ 20° was higher in the case of starches of WT and *su* genotypes than for starches of *wx-*genotype (Figure 5, curve 2–5). Probably this could be accounted for by the formation of the amylose V_h_ ordered structure in amylose-rich starches [36,46,48]. The crystallinities *C* of starch samples calculated with Equation (1) are depicted in Table 2. According to these results, the crystallinity of a waxy starch (variety Mais agestano, 0% amylose content, wx genotype) was essentially higher (C = 41%) than for starch samples with 17–38% amylose content (varieties Luch, Baksanskaya sakharnaya, Gornaya chechenskaya, White Flint with genotypes *wt*, *su*, and *ae*, correspondently) (C = 26–30%). The higher crystallinity of *wx*-genotype starch was associated with the highest amount of long- and short-chain amylopectin in this starch compared to starches of other genotypes [11]. Considering the main contribution to the crystalline structure of starch granules determined by amylopectin molecules and the higher crystallinity of *wx*-starch, this starch would melt at a higher melting temperature. The crystallinity of *su*-genotype starch was less than for other investigated starches (Table 2). Probably this starch contained the least amount of short- and long-chain amylopectin and would melt at a lower temperature than starches of *wt* and *wx* genotypes.

### 3.4. Thermal Properties of Starches in Different Maize Genotypes

Figure 6 presents examples of the DSC thermograms describing the melting of aqueous starch dispersions for various maize genotypes with different amylose content and genotypes.

The DSC thermograms are typical for melting aqueous dispersions of maize starches with different amylose content [37,41]. It is clear from the DSC thermograms that there are two peaks observed in the melting process of maize starch when starch contains a certain amount of amylose. The first peak corresponded to the melting of amylopectin crystalline lamellae or destruction (untwisting) of the double helices in amylopectin, while the second one marked the dissociation (melting) of amylose–lipid complexes [36,38,41,43]. Naturally, only one peak was observed while melting the amylopectin (*wx-*genotype) maize starch extracted from varieties Mais agestano and Mestnaya because this starch was amylose-free. Table 3 shows that the melting temperature values for amylopectin crystalline lamellae in the investigated starches decreased from 345.8 K (variety Mais agestano) to 340.8 K (variety Baksanskaya Sakharnaya), which is probably due to an intensified accumulation of defects in partially crystalline starch granules with an increase in their amylose content and differences in the granular structure of investigated starches. An important parameter that measures the energy required for the dissociation of the molecular double-helix order is the enthalpy of melting. The melting enthalpy values decreased with an increase in amylose content, explained by the reduction in the degree of crystallinity, which makes starch granules less resistant to gelatinization. As shown, starches with the long-branch chain length of amylopectin had higher values of melting enthalpies, which required more energy to melt crystallites with a long chain length [6,12]. The investigated maize starches likely differed in the long-branch chain length of amylopectin but supported the request for additional investigation. The highest values of melting temperature and enthalpy for amylopectin crystalline lamellae were observed in amylopectin starches as compared to normal maize starches (15–30% of amylose content), suggesting that amylopectin starches require more energy to disrupt their structure and that these starches have a more ordered crystalline structure and, correspondingly, the highest structural rigidity compared with other investigated maize starches [12,36,41]. With this, starch melting enthalpy values in the transitional from *wx* to “*ae”* genotype decreased with an increase in amylose content, while in the “*ae”-*genotype starches with the amylose content of 32.0 and 38.0%, the amylopectin melting enthalpy values virtually matched the respective values in amylopectin starches from the *wx* genotype. The higher transition temperature is associated with higher crystallinity with provided structural stability [6,12]. The starches of the *wx* genotype (varieties Mais agestano and Mestnaya) showed the highest value of the melting temperature among the investigated maize starches (Table 2), which may be connected with structural rigidity and more resistant to gelatinization, while starches of su-genotype (varieties Baksanskaya saharnaya, Alina, Ranyaya Lacomka) showed the lowest. The differences (ΔT) between conclusion temperature (T_c_) and onset temperature (T_0_) of amylopectin crystalline lamellar melting for investigated maize starches may be due to the changes in the crystalline regions in starch granules [40]. The higher value of ΔT for the *su*-genotype (varieties Baksanskaya sakharnaya, Alina, Ranyaya Lacomka) among investigated maize starches (Table 3) showed that these starches had lower crystallinity and the presence of crystalline regions of different strengths in starch granules correlated with our XRD results and literature data [11].

As shown earlier, during heating in excess of salt solution, the temperature transition of starches shifts to a higher value; at least the differences for A-type polymorphous structure correspond to 6–12 degrees; for B-type of polymorphous structure, the shift for melting temperature in excess of salt solution is 2–4 K [49]. The melting temperature of the investigated aqueous starch dispersions and starch dispersions in the presence of 1.6 M KCl solution showed that the melting temperatures of starches suspended among KCl were up to 6–8 K higher (Figure 6, dotted lines), which is typical for A-type polymorph and correlated with XRD results, so the type of polymorphic structure did not depend on the plant’s genotype in our study. 

The melting enthalpy value of amylose–lipid complexes is determined by the crystalline state of amylose–lipid complexes and is proportional to the content of lipids in them [50]. Note that the temperature and melting enthalpy values for amylose–lipid complexes in the investigated maize starches were practically unaffected by an increase in amylose content and remained invariable. Therefore, the lipid content and the crystalline state of amylose–lipid complexes in the investigated maize starches were practically the same.

Thus, the analysis of the studied thermodynamic parameters in the maize starch melting patterns of different genotypes with varying amylose content showed that an increase in amylose content in starch was accompanied by a decrease in amylopectin melting temperature values and determined by the genotypic features of the plant whose grain used for starch extraction (Table 3), which is in line with published data [41,51]. As shown, the formation of more defective (less-ordered) crystalline structures, melting at lower melting temperatures, accompanied an increase in amylose content in starches. The development of more defective (less-ordered) structures occurs because of the accumulation of non-ordered amylose chains in amorphous lamellae and amylose terminal chains in crystalline lamellae with increasing amylose content in starches [41,51] with the same pattern observed in the studied maize starches as well. As shown earlier, DSC data can apply to estimate the thickness of crystalline lamellae, and these data correlate well with data from XRD-analysis, for example [52,53]. The values of crystalline lamella thickness, evaluated from DCS data, changed from 4.7 to 7.4 nm for all tested starches (Table 3). In the case of *wx*-genotype starches, the value of the crystalline lamellae thickness was the lowest among investigated starches because *wx*-genotype starches had the lowest content of defects in the structural organization of starch granules. An increase in the number of defects, for example, such as amylose tie-chains, in starches of other studied genotypes accompanied an increase in the value of crystalline lamellae thickness.

It is worth mentioning that the thermodynamic melting parameters of the *su*-type maize starch extracted from Baksanskaya sakharnaya, Alina, and Ranyaya Lacomka, to some extent, fell out of the line of the tested starches. The melting of *su*-genotype starches had the lowest temperature and melting enthalpy among investigated starches, with the highest content of less-ordered structures, compared to starches of other studied genotypes (Table 3). Simultaneously, these starches demonstrated a large amount and the highest thermostability of their amylose–lipid complexes. As shown, maize starches from plants of *su*-genotype are characterized by the presence of V_h_-crystalline patterns that are attributes of amylose–lipid complexes [54]. Specific features of the starches extracted from the maize of *su*-genotype seem to be associated with the genotypic peculiarities of the sugary maize grain endosperm, characteristic of such cultivars, i.e., the effects of the *su*2 gene on starch granule development in the cultivar’s grain (the *su* genotype) are expressed in the form of disturbances in the starch polysaccharide branching process, as opposed to the other tested starches from the *wx* and “*ae*” genotypes. This fact, however, requires further and more profound research.

As known, the melting of polymer crystals begins with the melting of defects in an amorphous lamella. Here, amylopectin B-chains and amylose “tie chains” located in amorphous lamellae defects play the role of such defects [36,44]. The quantity of defects in the structural arrangement of starch granules is proportional to the value of surface energy on the terminal faces of a crystalline lamella of starch granules [44,55]. Qualitative evaluation of defects is possible using the method of differential scanning microcalorimetry.

The accumulation of defects accompanies the development of crystallites with a more ‘mellow’ surface. Thermodynamic parameters were assessed for the surface of the faces of crystalline lamellae in the tested starches (Table 4). The calculations involved the values of T_0m_ (366.5 K) and ΔH (35.5 J/g) as well as p_crl_ (1.48 g/cm^3^) for the A-type crystalline spherulites to which maize starches are attributed [56]. The values of melting entropy, melting temperature, and crystalline lamella thickness were the experimental ones. It is evident from the presented data (Table 4) that the surface entropy value for the faces of crystalline lamellae slightly increased for starches extracted from WT and “*ae*” genotypes compared to the corresponding value for starches from *wx*-genotype. This phenomenon indicated that defects in the starch structure accumulated with increasing amylose content in maize starch of the transition from *wx* to “*ae*”-genotype. It is clear from the presented data (Table 4) that the highest values of the surface entropy of the ends of crystalline lamellae were characterized by WT genotype starches, as well as the starches isolated from *su*-genotypes (Alina and Rannaya Lacomka varieties), probably due to their less-ordered structure compared to other starches studied. Note that for starches of the “*ae*” genotype with an amylose content of 32.0% and 38.0%, the value of the entropy of the end faces was close to the corresponding value for starches of the WT genotype with an amylose content of 15% and the *su*-genotype of the *Baksanskaya sakharnaya* variety, which probably indicated about the similarity in the degree of their order. It should be noted that starches with ploidy equal to four (varieties Populyatsiya MRPP22 and Baksanskaya sakharnaya) were characterized by closely free surface entropy values. It is likely that these starches were characterized by the same number of less-ordered structures. It can be assumed that the number of genes of diploid and tetraploid genotypes also determines the thermodynamic and structural features of starches, but this fact requires further research.

Thus, an increase in the amylose content in the tested maize starches accompanied a decrease in their thermodynamic melting parameters; simultaneously, the amount of amylose–lipid complexes remained practically unchanged for starches from WT and “*ae*” genotypes. The amylose content and the genotype of plants used for extracting starches determined changes in the thermodynamic parameters of investigated maize starches.

## 4. Conclusions

The studied high-starch maize accessions contained different starch genotypes with variable amylose/amylopectin ratios. Among the starch types of maize studied, four groups comprised the *wx*, “*ae*”, *su*, and transitional from *wx* to “*ae*” starch genotypes. The samples of maize varieties Baksanskaya sugar (*su*-genotype) and Population MRPP22 (transitional from *wx* to ”*ae*” genotype) had a tetraploid genotype but did not show signs of a pronounced specific effect of genome ploidy on the composition or ratio of amylose/amylopectin.

As shown, starch granules of the “*ae*” genotype had the highest values in granulometric sizes and from *su*-genotype—the minimum granulometric sizes among the tested starches. An increase in the amylose content in investigated maize starches with different genotypes accompanied a decrease in their thermodynamic melting parameters and the accumulation of defective structures in the studied starches, regardless of their ploidy. The starches extracted from the *su*-genotype differed in their thermodynamic melting parameters from other accessions having lower crystallinity and the highest content of less-ordered structures. Thus, amylose content in starch and the individual features of the maize genotype determine the thermodynamic melting parameters of studied starches. Probably the ploidy of maize determines the physicochemical properties of the extracted starches, but this requires further research.

The studied high-starch waxy maize accessions with the *wx* genotype (varieties Mais agestano, Mestnaya), with wild genotype maize accessions (varieties Luch and Kabardinskaya belaya zubovidnaya), and accession with the *su* genotype (varieties Baksanskaya sakharnaya, Alina, Ranyaya Lacomka) are of practical importance: they may serve as valuable raw materials for starch and treacle production industries and as sources for the development of new high-starch cultivars of waxy maize and traditional grain cultivars with a high mass fraction of starch. 

A comparative analysis of tetraploid and diploid samples differing only in the presence of one allele of the sugar content gene in the dominant and recessive states clearly showed a significant effect of the *su* gene on the phenotypic, morphological, and thermodynamic characteristics of native maize starch. Our study shows that genes involved in maize starch biosynthesis have a multifactorial effect, which manifests itself as a regulation of the parameters of thermodynamic and structural features of maize starch and biochemical ones of grain. These features may change more under the influence of several different in their effects genes than their number. Breeding and genetic engineering techniques make it possible to obtain starch with a wide range of physicochemical properties and starch grain phenotypes due to various combinations of several genes (*wx*, *ae*, *su*) in the genome of a diploid or tetraploid maize plant affecting starch synthesis. The study of the features of various maize genotypes, deciphering their genetic components, and their effects affecting the properties of starch will significantly improve the technology for producing cereal starches with specified physical and chemical properties as raw materials for the relevant industries.

## Figures and Tables

**Figure 1 polymers-15-01976-f001:**
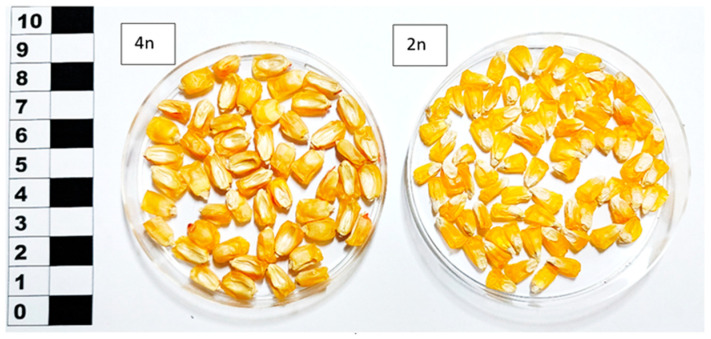
The grain size of tetraploid sugar maize variety Baksanskaya sakharnaya (4*n*) compared with diploid sugar maize variety Alina (2*n*).

**Figure 2 polymers-15-01976-f002:**
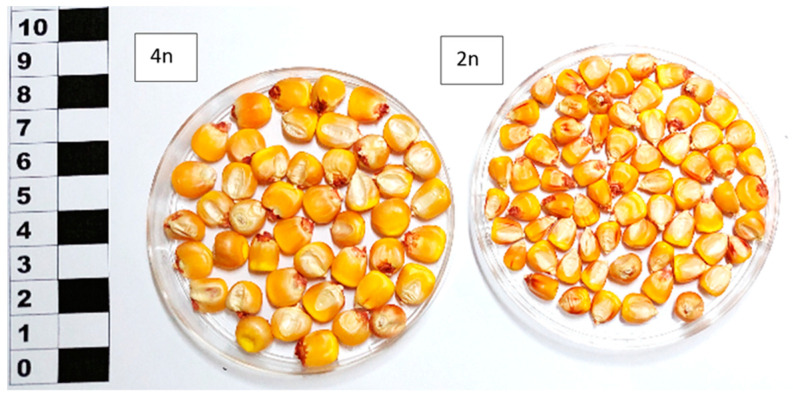
The grain size of tetraploid maize variety Population MRPP22 (4*n*) compared with diploid maize variety Luch (2*n*).

**Figure 3 polymers-15-01976-f003:**
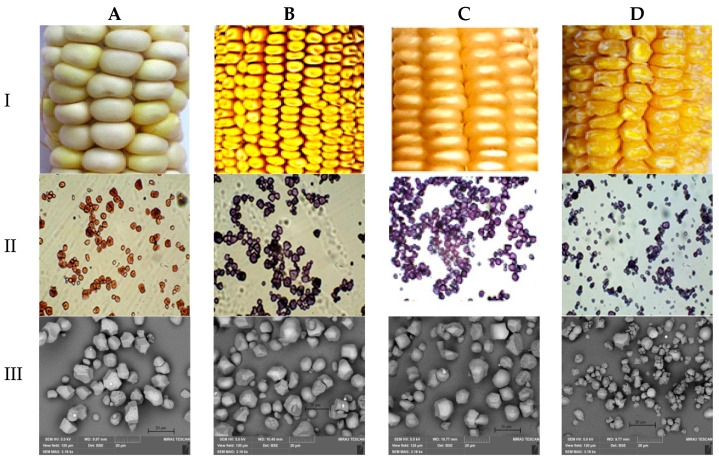
Examples of images of phenotypic trains in maize grain (**I**), as well as light (**II**) and electron scanning (bar = 20 µm) (**III**) microscopies images of starch granules with different genotypes ((**A**) *wx*, (**B**) WT, (**C**) “*ae”*, (**D**) *su*). Light microscopy images of starch granules stained with Lugol’s solution with a 400× zoom.

**Figure 4 polymers-15-01976-f004:**
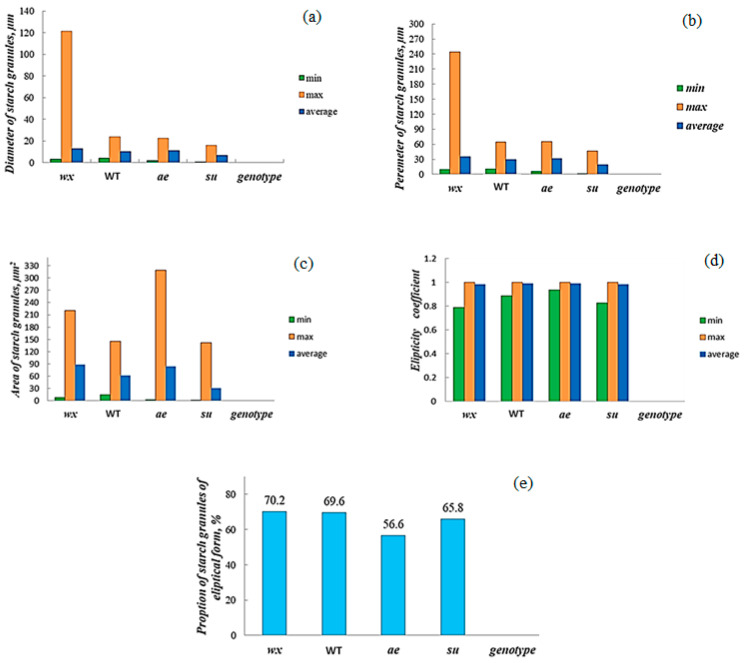
Examples of the granulometric data: diameter (**a**), perimeter (**b**), area (**c**), ellipticity coefficient (**d**), and proportion of elliptical forms (**e**) of starch granules from maize plants of different genotypes.

**Figure 5 polymers-15-01976-f005:**
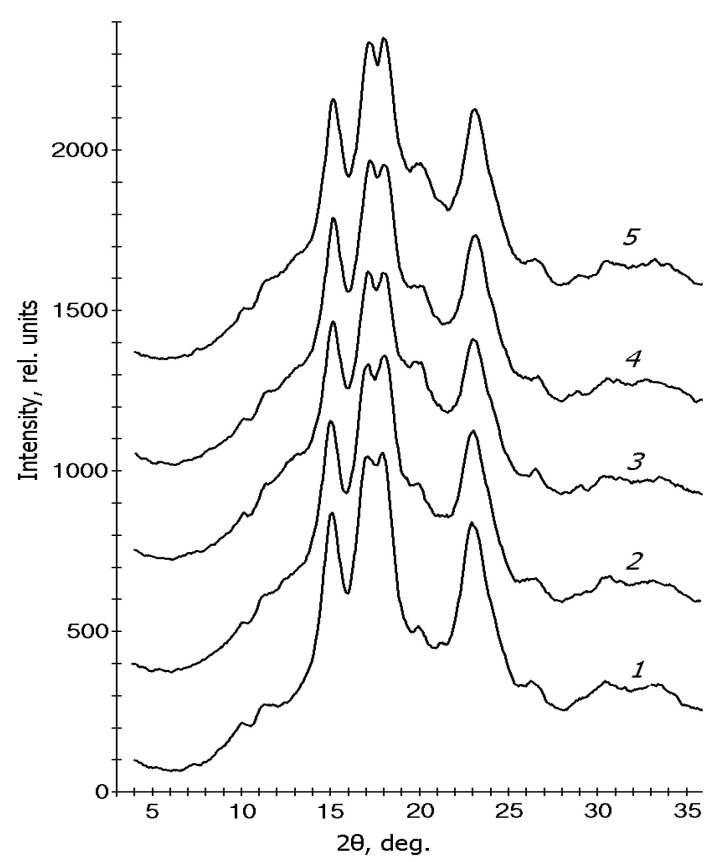
XRD patterns of starches of different maize genotypes and amylose content: 1—*wx* (0% the amylose content), 2—WT (17.5% amylose content), 3—*su* (25.5% amylose content), 4 and 5—“*ae*” (32% and 38% amylose content, correspondingly*)* genotypes.

**Figure 6 polymers-15-01976-f006:**
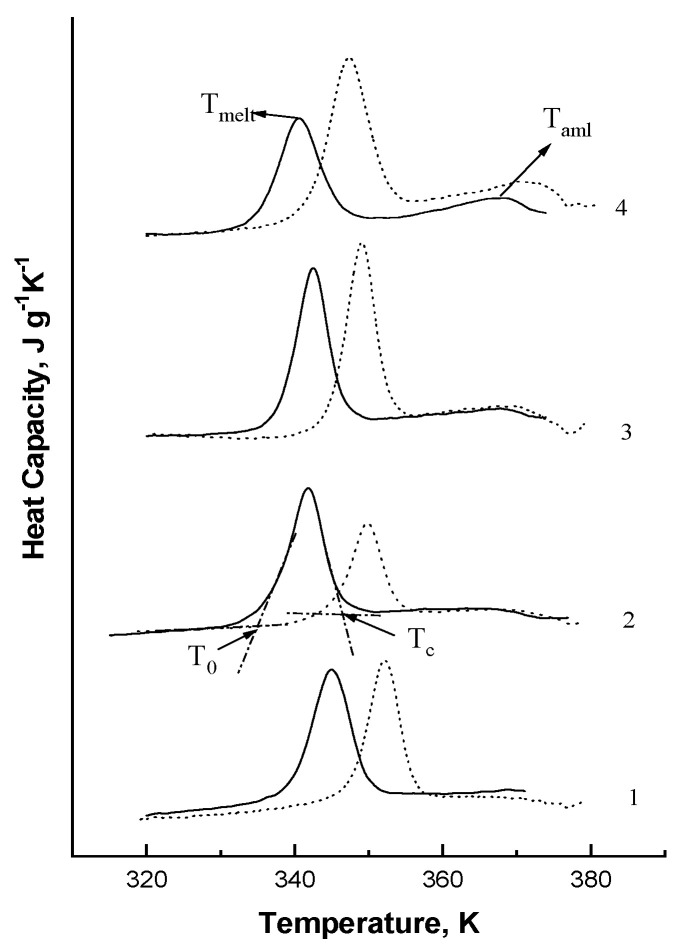
The examples of DSC thermograms of starch dispersions (0.3% *w*/*w*) in water (straight line) and in 1.6 M KCl (dotted line) with different genotypes and amylose content: 1—*wx* (0% amylose content), 2—wt (15% amylose content), 3—“*ae*” (38.0% amylose content), 4—*su* (25.5% amylose content) genotypes. T_m_ is the melting temperature for amylopectin crystalline lamellae; T_0_ and T_c_−onset and conclusion temperatures of amylopectin crystalline lamellae melting; T_aml_ is the melting temperature of the amylose–lipid complex.

**Table 1 polymers-15-01976-t001:** Values of the main components in the grain’s endosperm of different maize genotypes and effectiveness indicators of their processing into starch.

Cultivar	Ploidy, n	Genotype	Dry Matter Mass Fraction in Grain, %	Starch Content in Grain (S1), %	Mass Fraction of Ash Content in Grain, %	Amylose Content in Starch, %	Starch Yield after Maize Grain Processing under Laboratory Conditions (S2), % of Grain DM	Starch Extraction Ratio (S2/S1·100), %
Mais agestano	2	*wx*	92.2 ± 0.3	70.5 ± 0.1	1.7 ± 0.2	0.0 ± 0.5	58.6 ± 0.7	83.1
Mestnaya	2	*wx*	92.1 ± 0.4	70.8 ± 0.5	1.7 ±0.2	0.0 ± 0.5	59.9 ± 1.4	84.6
Populyatsiya MRPP22	4	WT	91.9 ± 0.4	73.7 ± 0.3	1.7 ± 0.1	15.0 ± 0.7	62.9 ± 0.9	85.3
Luch	2	WT	92.1 ± 0.2	75.3 ± 0.4	1.3 ± 0.1	17.5 ± 1.0	64.7 ± 0.7	85.9
Kabardinskaya belaya zubovidnaya	2	WT	91.4 ± 0.3	73.0 ± 0.1	1.5 ± 0.1	21.0 ± 0.9	59.9 ± 0.6	82.1
Otbornyy 150SV	2	WT	92.1 ± 0.2	70.9 ± 0.1	1.4 ± 0.1	26.4 ± 0.9	56.8 ± 0.4	80.1
Gornaya chechenskaya	2	“*ae*”	92.0 ± 0.3	71.3 ± 0.3	1.5 ±0.2	32.0 ± 0.9	62.0 ± 0.8	86.9
White Flint	2	“*ae*”	91.2 ± 0.5	69.3 ± 0.3	1.5 ± 0.1	38. 0 ± 0.1	62.6 ± 0.7	90.3
Baksanskaya sakharnaya	4	*su*	93.3 ± 0.3	59.3 ± 0.2	1.7 ± 0.1	25.5 ± 1.1	20.8 ± 1.4	35.1
Alina	2	*su*	92.8 ± 0.4	65.9 ± 0.6	1.4 ±0.2	26.4 ± 1.2	10.5 ± 2.3	15.9
Ranyaya Lacomka	2	*su*	93.3 ± 0.2	58.8 ± 0.4	1.5 ±0.1	27.7± 0.8	17.8 ± 2.1	30.3

**Table 2 polymers-15-01976-t002:** Crystallinities *C* of starch samples calculated on the basis of XRD data.

Cultivar	Genotype	Amylose Content, %	C, %
Mais agestano	*wx*	0	41
Luch	*WT*	17.5	29
Gornaya chechenskaya	“*ae*”	32.0	30
White Flint	“*ae*”	38.0	30
Baksanskaya sakharnaya	*su*	25.5	26

**Table 3 polymers-15-01976-t003:** Thermodynamic melting parameters of maize starches with different genotype (melting temperature (T_m_), melting enthalpy (ΔH_m_), van’t Hoff’s enthalpy(ΔH^vH^), cooperative melting units (ν), and thickness (L_crl_) of crystalline lamellae; T_aml_ and ΔH*_a_*_ml_ are the temperature and enthalpy of amylose–lipid complex melting).

Cultivar	Genotype/ Ploidy, n	T_m_, K	ΔT = Tc ࢤ T_0_	ΔH_m_, kJ/mol	ΔH^vH^, kJ/mol	ν, Anhydroglucose Residue	L*_crl_*, nm	T_aml_, K	ΔH_aml_, kJ/mol
Mais agestano	*wx/2*	345.8 ± 0.1	9.7	3.7 ± 0.2	48.6 ± 1.5	13.4 ± 0.7	4.7 ± 0.2	-	-
Mestnaya	*wx/2*	345.2 ± 0.1	10.0	2.7 ± 0.2	38.1 ± 1.5	14.2 ± 0.6	5.0 ± 0.2	-	-
Populyatsiya MRPP22	WT*/4*	342.1 ± 0.1	10.7	2.4 ± 0.1	38.2 ± 0.1	16.0 ± 0.5	5.6 ± 0.2	366.5 ± 0.1	0.2 ± 0.05
Luch	WT*/2*	343.7 ± 0.1	8.4	1.9 ± 0.1	34.9 ± 1.8	17.9 ± 0.1	6.3 ± 0.0	365.6 ± 0.0	0.30 ± 0.05
Kabardinskaya belaya zubovidnaya	WT*/2*	343.1 ± 0.0	8.6	1.9 ± 0.3	36.2 ± 4.0	20.0 ± 0.8	7.0 ± 0.3	365.7 ± 0.1	0.10 ± 0.0
Otbornyy 150SV	WT*/2*	343.2 ± 0.1	7.3	2.1 ± 0.3	42.5 ± 2.4	21.0 ± 3.0	7.4 ± 0.7	366.1 ± 0.0	0.30 ± 0.05
Gornaya chechenskaya	“*ae*”*/2*	343.5 ± 0.0	10.2	2.8 ± 0.3	43.2 ± 2.5	15.5 ± 0.8	5.5 ± 0.2	366.0 ± 0.0	0.30 ± 0.03
White Flint	“*ae*”*/2*	342.9 ± 0.0	10.4	2.7 ± 0.1	42.1 ± 1.2	16.0 ± 0.4	5.6 ± 0.1	365.8 ± 0.0	0.40 ± 0.0
Baksanskaya sakharnaya	*su/4*	340.8 ± 0.1	12.3	2.4 ± 0.1	35.1 ± 0.4	14.7 ± 0.8	5.2 ± 0.2	367.3 ± 0.0	0.7 ± 0.1
Alina	*su/2*	342.1 ± 0.1	12.1	1.6 ± 0.1	27.9 ± 1.2	16.9 ± 0.4	6.0 ± 0.1	367.5 ± 0.1	0.3 ± 0.05
Ranyaya Lacomka	*su/2*	341.8 ± 0.1	14.5	1.8 ± 0.1	28.5 ± 0.8	15.8 ± 0.4	5.5 ± 0.2	366.6 ± 0.1	0.5 ± 0.1
Mean value								365.9 ± 0.7	0.4 ± 0.1

**Table 4 polymers-15-01976-t004:** The values of free surface entropy (s_i_) on the face side of crystalline lamellae in maize starches with different genotypes.

Cultivar	Genotype/ Ploidy, n	s_i_ × 10^7^ J/cm^2^
Mais agestano	*wx/2*	0.08
Mestnaya	*wx/2*	0.13
Populyatsiya MRPP22	WT*/4*	0.17
Luch	WT*/2*	0.23
Kabardinskaya belaya zubovidnaya	WT*/2*	0.26
Otbornyy 150SV	WT*/2*	0.26
Gornaya chechenskaya	“*ae*”*/2*	0.15
White Flint	“*ae*”*/2*	0.16
Baksanskaya sakharnaya	*su/4*	0.16
Alina	*su/2*	0.23
Ranyaya Lacomka	*su/2*	0.20

## Data Availability

Data is available on request.

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
