# Peer review of "Main Characteristics of Processed Grain Starch Products and Physicochemical Features of the Starches from Maize (Zea mays L.) with Different Genotypes"

_polymers, 2023, doi:10.3390/polym15081976_

Round 1
Reviewer 1 Report (Previous Reviewer 1)
The scientific contribution of this article to the is extermely low. The genotypes investigated are known for decades in the case of maize and the resulting starches have been much better characterised in the past in many publications.
In this work the characeterisation of the satrch is incomplete and provide less information than older works.
This is not possible anymore to accept a publication about starch characterisation with DSC analysis as main and almost unique characterisation technique, even if the treatment of the DSC results seem correct.
Besides the answer to the reviewer remark "The starch structure characterisation is too limited. DSC is fine but one would except chain length distribution, NMR or X-ray analysis and so on, for native starch characterization" is incorrect.
The authors justify that no other techniques were used with this new text : « It is known that maize starches are characterized by A-polymorphous structure "
This is an obvious oversimplification, starch from wild-type maize does present A-type crystallinity, but starch from ae genotype, such as the one investigated in this work, usually exhibits B-type crystallinity. This is common knowledge and a basic characterisation to determine the polymorphic type is X-ray analysis. If the authors cannot even confirm the dominant crystalline polymorphic type of their samples then the work is simply not qualified to be published in this journal.
Author Response
The authors are grateful to the referee and accept all comments. The manuscript has been corrected in accordance with all the comments of the reviewers (highlighted with a yellow marker).

Reviewer 2 Report (New Reviewer)
The Authors report an interesting study of the relationship between the genotype of maize plants, the thermodynamic and morphological features of starches from the grains of these plants. Using light and scanning electron microscopy, they obtained granulometric data of extracted starches and from differential scanning calorimetry they obtained thermodynamic melting parameters. Authors discuss possible connections between amylose content and different experimentaly obtained structural and thermodynamic parameters. The manuscript reports some interesting findings, can help in obtaining starches with desired physical and chemical properties as raw materials for the relevant industries, it is well-written but there are several major issues/comments to consider before publication. These comments can be found as comments in the attached file.
Also, the authors have included a lot of experimental data and calculated parameters but not all of them have been explained or utilised to support manuscrip's conclusions. Is it really necessary to report all that data? Perhaps some could be omitted thus shifting the focus to more important ones. Data from different experimental techniques show that amylose content is not the only thing that influences certain parameters (melting enthalpy, free surface entropy, ...). Results from different experimental techiques are explained induvidually and I think more effort should be made to look for correlations between them.

Author Response
The authors are grateful to the referee and accept all comments. The manuscript has been corrected in accordance with all the comments of the reviewers (highlighted with a yellow marker).

Reviewer 3 Report (New Reviewer)
1. Please include a couple of lines about ploidy levels in maize in general as you have used tetraploid genotypes also in the study.
2. Smiliar kind of work if done in context to other starch from different crops and practical significance need to be incorporated.
Author Response
The authors are grateful to the referee and accept all comments. The manuscript has been corrected in accordance with all the comments of the reviewers (highlighted with a yellow marker).

Reviewer 4 Report (New Reviewer)
This paper (polymers-2160504) aims to study main characteristics of grain processing products on starch and physicochemical features of the starches from maize (Zea mays L.) with different genotypes. And the experimental results are discussed in detail, including the peculiarities of starch extracted from subspecies of maize (the dry matter mass (DM) fraction, starch content in grain DM, ash content in grain DM, and amylose content in starch). I think major revision is needed. The following questions to be considered may be helpful to improve the manuscript.1. Abstract: what do you mean wx, “ae”, su?
2. Abstract: The description transitional from wx to “ae” genotypes is not a scientific term.
3. In this study, the authors propose that an increase in amylose content increases defects in partially crystalline starch granules that decrease starch melting temperature. However, high amylose maize starch resist to be gelatinized, how do you explain this phenomenon?
4. The physicochemical features of the starches from maize with different genotypes was investigated, how did you make sure the materials used in this study is proper to reflect the main characteristics of grain?
5. Table 2 was shown with incomplete information. Please revise.
6. Too many References were shown in this manuscript. Please revise.
Author Response
Response to Reviewer 4:
1.You write: Abstract: what do you mean wx, “ae”, su
Our answer: Four groups comprised the waxy (wx), high amylose (“ae”), sugar (su) and wild(wt) genotypes were investigated in this study. Starches with an amylose content of over 30% conditionally belonged to the “ae” genotype.
- You write: Abstract: The description transitional from wx to “ae” genotypes is not a scientific term.
Our answer: we agree with this remark. Throughout the text, we have replaced “transitional from wx to “ae” genotypes” on “wild (wt) genotype”
- You write: In this study, the authors propose that an increase in amylose content increases defects in partially crystalline starch granules that decrease starch melting temperature. However, high amylose maize starch resist to be gelatinized, how do you explain this phenomenon?
Our answer: High amylose starches are generally considered to contain more than 50% amylose.
It is known that high-amylose corn starches are characterized by higher melting temperatures, which is associated with a change in the polymorphic structure from A type for normal starches to a mixed B type and B * for high amylose starches , with a thickness of crystalline lamella. Maize samples with an increased amylose content of more than 30% ( varieties White Flint, Gornaya chechenskaya) in our study are conditionally assigned to subspecies of amylose type "ae" since the increased content of amylose starch does not allow us to classify these samples as normal, but they are not yet typically high amylose.
- The physicochemical features of the starches from maize with different genotypes was investigated, how did you make sure the materials used in this study is proper to reflect the main characteristics of grain?
Our answer: To assess the use of these starches, it is necessary to understand their physicochemical properties and also to know the processing characteristics of a given variety of a particular raw material for starch and its processed products, which we tried to evaluate in this study. The samples used as a model in the studies were studied to determine their polymorphism according to economically valuable traits of the world collection of VIR corn. These samples are represented by a food subspecies widely used and popular for human nutrition. Therefore, the authors focused the research area on the description of the main (standard) qualities of the starch of these samples.
You write:5. Table 2 was shown with incomplete information. Please revise.
Our answer: Сolumn 2 added to table 2
You write: 6. Too many References were shown in this manuscript. Please revise.
Our answer: We done it and decreases references.

Reviewer 5 Report (New Reviewer)
The comments are in the file attached

Author Response
Response to Reviewer5:
You write: Can you have tetraploid maize of the waxy and high amylose types? If so, why were they not included in the study? Since only 2n and 4n versions are included for sweet corn.
Our answer: In this collection, exactly those varieties that we tried to study were found. We do not have other samples yet, perhaps in the future we will continue research
You write: how is grain yield affected by having them in the 4n version?
Our answer: Tetraploid varieties differ from diploid varieties in larger grain and embryo sizes. This allows you to increase the yield of starch from the grain.
You write: In order to facilitate the understanding of this section, and the aims of the study could be clearer, it is suggested that authors re-write it.
Our answer: we tried to do it
You write: Why you did not quantify the lipid content in the starches obtained? Taking in account the importance of lipids on the physicochemical properties of starches.
Our answer: We agree that the lipid content in starches is related to their physicochemical properties, we have shown by DSC that the enthalpy of fusion of amylose-lipid complexes is slightly different in the studied starches, which is due to the amount of lipids in them. Thus, we tried to estimate the amount of lipids in starches.
You write: Why did you not add information about the genotype of several cultivars in Table 1?
Our Answer: It done (see Table 1)
Please, check well the redaction of paragraph “Studying the ranking………Figure 1 (I). Because the information regarding the ploidy of the cultivars was in Table 1 and not in Table 2.
You write: Subsection 3.1 and 3.2 could be combined in one section, because both refer to information of Table 1.
Our Answer: It done
You write: What is the importance of determine ash content in the starch?, by the relation with the protein content?
Our Answer: Table 1 shows the ash content of grain, not starch. The ash content in the grain is related to the amount of mineral fertilizers used in the cultivation of maize plants.
You write: More discussion of the results are necessary. For example, the size of the starch granules in waxy (wx) has been previously reported ( ), however no information is available about the size of starch granules of wx maize 4 n.
Our Answer: we tried to do this and changed the presentation of granulometric data
You write: The edition of the title of tables and figures is recommended. The titles must be explicative by themselves.
Our Answer: We tried to improve the titles for figures and table

Round 2
Reviewer 1 Report (Previous Reviewer 1)
This is a resubmission of a manuscript for which I suggested the rejection for publication. This version of the manuscript now introduces an research question, the potential effect of increasing the ploidy of maize subspecies genome on starch properties, which is admitedly a significant improvement compared to the initial version. Unfortunetly the study was not originally designed to address this topic and the impact of the "ploidy" on the starch samples is not visible in most graphs. Besides the sudy still suffers from inadequate and insufficient characterisation of the starch materials.
As for illustration, one can consider Figure 2 which reports SEM based data about starch granule sizes.
It presents only minimum and maximum values, but for perimeter, area and the % or elliptical starch granule while the obvious information to give is the median or average value of area or, as often in publications, the estimated granules diameter. Ideally the full size distribution or a boxplot would be supplied because the min and max values cannot not convey the important information to the reader. Besides tracking the % of irregular granules through the analysis of the % of elliptical granules is not very convincing and certainly not vital for the reader. Finally it is unlikely that a SEM based analysis can be representative of the actual distribution of starch granules size. Usually a laser diffraction instrument is used for this.
From Fig.3 it can be seen that all samples investigated present A-type crystallinity pattern. Why did the authors not estimated the % of crystallinity to better discriminated their samples ? Again this is a basic analysis provided in most publications on this topic, that the reader would expect.
Finally a thorough DSC-based analysis is supplied. While it is scientifcally correct, it only confirms well established knowledge about the impact of amylose content on crystallinity reflected by the thermograms.
To me this manuscript is not adequate materials for publication in a scientific journal.
Author Response
Response to Reviewer 1:
The samples used as a model in the studies were studied to determine their polymorphism according to economically valuable traits of the world collection of VIR corn. These samples are represented by a food subspecies widely used and popular for human nutrition. Therefore, the authors focused the research area on the description of the main (standard) qualities of the starch of these samples. In our opinion, rather interesting starches were found from this collection, characterized by different ploidy. We tried to characterize these starches of different genotypes using the methods available to us. We agree that we did not quite correctly present the results of the SEM study of our starches. In this version, we tried to take into account your remark (Figure 4). The value of crystallinity of starches of various genotypes is also given (Table 2).

Reviewer 2 Report (New Reviewer)
The authors have adressed most of my concerns. The manuscript has been significantly improved. The only remaining concern I have with the manuscript is that symbols representing physical quantities (or variables) should be written in italic.
Author Response
Response to Reviewer 2:
You write: The authors have adressed most of my concerns. The manuscript has been significantly improved. The only remaining concern I have with the manuscript is that symbols representing physical quantities (or variables) should be written in italic.
Our answer: It done.

Reviewer 4 Report (New Reviewer)
Minor spell check is required before publication.
Reviewer 5 Report (New Reviewer)
Please check carefuly the use of "maize" and "corn", In the document "maize" predominates, however, one "corn" was still detected in the Introduction section.
Define "VIR" abreviation the first time you mentions it in the document
It is not clear the idea in the last paragraph of "Conclussions" section. In the row "influence of several different in their effects genes than their number". Please check.
This manuscript is a resubmission of an earlier submission. The following is a list of the peer review reports and author responses from that submission.
Round 1
Reviewer 1 Report
As ackowledged by authors of this paper in their previous articles "the melting thermodynamic properties of starches with differences in amylose content have been studied intensively". And starch from maize mutants are probably the most studied materials. Therefore, it is very challenging to make an original contribution on this topic.
While the article presents good points, original maize materials with potentially interesting ultrastructure (unfortunately not sufficienty characterised), detailed DSC analysis, it is lacking by too many aspects to present any significant interest for domain researchers :
- The starch structure characterisation is too limited. DSC is fine but one would except chain length distribution, NMR or X-ray analysis and so on, for native starch characterization.
- Results are poorly presented. Pictures of Fig1 are too small, Fig1 and Fig2 present four type of starch, while eleven cultivars are analysed in total. No distribution of the granule size is provided as graphs to illustrate the long text 3.3. Analysis of the results is sometimes questionable e.g. l.341, "it is obvious from the DSC thermograms that there are two peaks.." while the second peak is barely visible.
- Results are not sufficiently analysed. No stat analysis to relate genotype with structure or DSC results.
- Resulst are not sufficiently discussed with regard to existing publications, which is really damaging for the paper interest. This would allow to highlight new findings. In its current state this is just one more paper about starch characterisation.
- Blibliography is clearly imcomplete, some important articles about maize starch from Inouchi, Gérard, Dhital and so on are not cited. There are several review papers about starch characterisation published quite recently that can be helpful.
Author Response
Response to the Editor and Reviewers
First of all the authors would like to thank you for your benevolent attitude to our manuscript. We are also thankful to you for attentive reviewing our article and for the valuable remarks made. The manuscript has sufficiently been modified and some text have been added in accordance with your remarks. We have made necessary amendments to the text, figures, table, and tried our best to clarify the language. We strongly hope that the changes addressed those issues or shortcomings of the manuscript, which the editor and reviewers made. Responding to your remarks, we would like to report the following:
Response to Reviewer 1:
You write: - The starch structure characterisation is too limited. DSC is fine but one would except chain length distribution, NMR or X-ray analysis and so on, for native starch characterization.
Thank you for your remark, but we shown by DSC that the polymorphous structure does not changed and buy our opinion this is enough for understanding the type of polymorphous structure, in the next publications we’ll take into account your remark. Below there is the part about polymorphous structure of the investigated starches: “It was shown that the polymorphic structure of these starches does not change « It is known that maize starches are characterized by A-polymorphous structure [N. Singh, N. Inouchi, K. Nishinari, Structural, thermal and viscoelastic characteristics of starches separated from normal, sugary and waxy maize, Food Hydrocoll. 2006, 20, 923–935. [DOI 10.1016 / j.foodhyd.2005.09.009]]. It was shown earlier that during heating in an excess of salt solution the temperature transition of starches shifts to a higher value, at least the differences for A-type of polymorphous structure correspond to 6–12 degree [T. Y. Bogracheva, V.J. Morris, S.G. Ring, C.L. Hedley: The granular structure of C-type starch and its role in gelatinisation. Biopolym. 1998, 45, 323–332]. The melting temperature of the investigated aqueous starch dispersions and starch dispersions in the presence of 0.6 M KCl solution shows that the melting temperatures of starches suspended in KCl are up to 6–7 K higher (Figs. 2, 1a). This is typical for A-type polymorph, so the type of polymorphic structure doesn’t depend on the plants genotype.”
You write:- Results are poorly presented. Pictures of Fig1 are too small, Fig1 and Fig2 present four type of starch, while eleven cultivars are analysed in total. No distribution of the granule size is provided as graphs to illustrate the long text 3.3. Analysis of the results is sometimes questionable e.g. l.341, "it is obvious from the DSC thermograms that there are two peaks.." while the second peak is barely visible.
We gave examples of photos and DSC thermograms, in our opinion it does not make sense to give all DSC thermograms, they are all typical, we tried to improve the drawing with DSC thermograms and discuss the results in more details
You write:- Results are not sufficiently analysed. No stat analysis to relate genotype with structure or DSC results.
Statistical analysis done and shown in Table
You write:- Resulst are not sufficiently discussed with regard to existing publications, which is really damaging for the paper interest. This would allow to highlight new findings. In its current state this is just one more paper about starch characterisation.
- Blibliography is clearly imcomplete, some important articles about maize starch from Inouchi, Gérard, Dhital and so on are not cited. There are several review papers about starch characterisation published quite recently that can be helpful.
We tried to improve the discussion of the results and expand the bibliography in accordance with the publications found

Reviewer 2 Report
The study here is not very innovative and only few data is available. The way that the results is not complete and clear. The language needs to be improved since there are many grammar errors (see Line 16-17, 123, 294, 311, 366, 374-375). There are many issues that need to be addressed.
1. Pay attention to the lower letter of preposition in the title.
2. The title mentions the “Characteristics of The Main Products of Maize Grain Processing with wx, ae And su Mutations in The Genome of Endosperm 3 (Zea mays L.) for Starch” but there limited data exists concerning the characteristics of the products, and most data is for the starch in this article.
3. The abstract given here starts without any background of the present work. The purpose of the study undertaken and the important conclusion based on the obtained results should be briefly stated.
4. The first sentence in abstract is too long, and the keywords need to be rewritten. Where is the data of rheological properties???
5. Line 59-62, “High-amylose maize starch is distinguished for such properties as increased solubility in water” High-amylose?? Not amylopectin?
6. Line 76. The full name of GT-B needs to be given when the abbreviation first appeared in the article.
7. The significant difference analysis is missed in Table 1.
8. Line 301. Ensure to unity the figure description in the article.
9. Line 304-315. Please show the data of granule sizes in a table.
10. Line 344 “while 343 the second one marked the dissociation (melting) of amylose–lipid complexes”. More relevant references need to be cited to support this conclusion.
11. Please unity the description of samples name in the manuscript.
12. Line 349. “Melting temperature values for amylopectin crystalline lamellae in the investigated starches decreased from 345.8 K to 340.8 K”. Please note the sample name here.
13. Line 357. “…….a more ordered crystalline structure compared to normal maize starches”. Add relevant references to support this conclusion.
14. Line 376. Change” in” to “among”.
15. The conclusion section is lengthy and redundant, and the research significance if this study needs to be clarified. Better to re-write this section.

Author Response
Response to the Editor and Reviewers
First of all the authors would like to thank you for your benevolent attitude to our manuscript. We are also thankful to you for attentive reviewing our article and for the valuable remarks made. The manuscript has sufficiently been modified and some text have been added in accordance with your remarks. We have made necessary amendments to the text, figures, table, and tried our best to clarify the language. We strongly hope that the changes addressed those issues or shortcomings of the manuscript, which the editor and reviewers made. Responding to your remarks, we would like to report the following:
Response to Reviewer 2:
- You write: Pay attention to the lower letter of preposition in the title.
The title mentions the “Characteristics of The Main Products of Maize Grain Processing with wx, ae nd su Mutations in The Genome of Endosperm 3 (Zea mays L.) for Starch” but there limited data exists concerning the characteristics of the products, and most data is for the starch in this article.
The title of the publication has been edited and now the title is following:
«Basic Characteristics of the Maize Grain Processing Products with wx, ae and su Mutations in The Genome of Endosperm (Zea mays L.) on Starch, as well as Morphological and Thermodynamic Features of The Isolated Starches»
- You write: The abstract given here starts without any background of the present work. The purpose of the study undertaken and the important conclusion based on the obtained results should be briefly stated.
We tried to take into account your comment and changed the abstract: «To understand the relationship between the genotype of maize plants, differing in their origin and the ploidy of the genome, which carry gene alleles programming the biosynthesis of various starch modifications, the thermodynamic and morphological features of starches from the grains of such plants have been studied. The peculiarities of the isolation of starch extracted from subspecies of maize (the dry matter (DM) mass fraction, starch content in DM grain, ash content in DM grain, amylose content in starch) belonging to different genotypes have been studied. Among the starch types of maize studied, four groups were earmarked to comprise the wx, ae, su and transitional starch types. The starches of the su genotype were shown to have the smallest starch granules compared to the wx and ae genotype starches. An increase in amylose content in the investigated starches was found to be accompanied by a decrease in their thermodynamic melting parameters, thus inducing the accumulation of defective structures in the studied starches. At the same time, the starches of ae and su genotypes possessed a more ordered structure than those of wx genotype. In the ae- and wx-type starches, thermodynamic parameters were evaluated for amylose–lipid complex dissociation, i. e. temperature (Taml) and enthalpy (Haml); in the case of the su genotype, temperature and enthalpy values of amylose–lipid complex dissociation were higher than those in the starches from the ae and wx genotypes. The thickness of the crystalline lamellae (Lcrl) for all studied genotypes was virtually independent of the amylose content or the starch genotype. It was shown that the thermodynamic melting parameters of studied starches are determined both by amylose content in starch and by the individual features of the maize genotype.»
- You write: The first sentence in abstract is too long, and the keywords need to be rewritten. Where is the data of rheological properties???
Rewritten , see abstract
5.You write :Line 59-62, “High-amylose maize starch is distinguished for such properties as increased solubility in water” High-amylose?? Not amylopectin?
This sentence has been deleted
- You write: Line 76. The full name of GT-B needs to be given when the abbreviation first appeared in the article.
The full name inserted (Glycosyltransferase-В (GT-B))
- You write: The significant difference analysis is missed in Table 1.
We added the description of Table 1 (see section 3.2)
- You write: Line 301. Ensure to unity the figure description in the article.
Done
- You write: Line 304-315. Please show the data of granule sizes in a table.
The Data of SEM analysis are shown on Fig.2
- You write: Line 344 “while 343 the second one marked the dissociation (melting) of amylose–lipid complexes”. More relevant references need to be cited to support this conclusion.
We are agree with this remark and now this part of the text is following:
«The observed DSC thermograms are typical for the melting of aqueous dispersions of maize starches with different amylose content [35, 39]. It is obvious from the DSC thermograms that there are two peaks observed in the melting process of maize starch when starch contains a certain amount of amylose. The first peak corresponds to the melting of amylopectin crystalline lamellae or destruction (untwisting) of the double helices in amylopectin, while the second one marked the dissociation (melting) of amylose–lipid complexes [35, 36, 37,39,47]. Naturally, only one peak was observed while melting the amylopectin (wx-genotype) maize starch extracted from varieties Mais agestano and Mestnaya, because this starch was amylose-free.»
- You write: Please unity the description of samples name in the manuscript.
Done
- You write: Line 349. “Melting temperature values for amylopectin crystalline lamellae in the investigated starches decreased from 345.8 K to 340.8 K”. Please note the sample name here.
We are agree with your remark and now this part of the text is following :«Melting temperature values for amylopectin crystalline lamellae in the investigated starches decreased from 345.8 K (variety Mais agestano) to 340.8 K (variety Baksanskaya Sakharnaya), which is probably due to an intensified accumulation of defects in partially crystalline starch granules with an increase in their amylose content.»
- You write: Line 357. “…….a more ordered crystalline structure compared to normal maize starches”. Add relevant references to support this conclusion.
According with your remark we insert relevant references (see the text)
- You write: Line 376. Change” in” to “among”.
We changed ” in” to “among”. And now the sentence is the following : The melting temperature of the investigated aqueous starch dispersions and starch dispersions in the presence of 0.6 M KCl solution shows that the melting temperatures of starches suspended among KCl are up to 6–7 K higher
- You write: The conclusion section is lengthy and redundant, and the research significance if this study needs to be clarified. Better to re-write this section.
We took into account your remark, and slightly reduced this part
